# Research Progress on the Mechanisms of Polysaccharides against Gastric Cancer

**DOI:** 10.3390/molecules27185828

**Published:** 2022-09-08

**Authors:** Liping Chen, Chunrong He, Min Zhou, Jiaying Long, Ling Li

**Affiliations:** 1School of Food and Bioengineering, Xihua University, Chengdu 610039, China; 2Key Laboratory of Drug Targeting and Drug Delivery Systems, Ministry of Education, West China School of Pharmacy, Sichuan University, Chengdu 610041, China

**Keywords:** anti-gastric cancer, polysaccharides, plants, fungi, mechanism

## Abstract

Gastric cancer is a common type of cancer that poses a serious threat to human health. Polysaccharides are important functional phytochemicals, and research shows that polysaccharides have good anti-gastric cancer effects. We collated all relevant literature published from 2000 to 2020 and found that more than 60 natural polysaccharides demonstrate anti-gastric cancer activity. At the present, the sources of these polysaccharides include fungi, algae, tea, *Astragalus membranaceus*, *Caulis Dendrobii*, and other foods and Chinese herbal medicines. By regulating various signaling pathways, including the PI3K/AKT, MAPK, Fas/FasL, Wnt/β-catenin, IGF-IR, and TGF-β signaling pathways, polysaccharides induce gastric cancer cell apoptosis, cause cell cycle arrest, and inhibit migration and invasion. In addition, polysaccharides can enhance the immune system and killing activity of immune cells in gastric cancer patients and rats. This comprehensive review covers the extraction, purification, structural characterization, and mechanism of plant and fungal polysaccharides against gastric cancer. We hope this review is helpful for researchers to design, research, and develop plant and fungal polysaccharides.

## 1. Introduction

Bray et al. evaluated global cancer incidence and mortality according to GLOBOCAN 2018, provided by the International Agency for Research on Cancer. Research showed that gastric cancer is still a serious threat to human health. There are more than one million new cases of gastric cancer and more than 700,000 deaths reported worldwide every year. Gastric cancer has become the fifth most frequently diagnosed cancer, and it is the third leading cause of cancer-related death [1]. At present, the clinical treatment of gastric cancer is largely chemotherapy, but these medicines cause adverse reactions. Fluorouracil can cause a loss of appetite, nausea, vomiting, and diarrhea [2]. Nitrosourea can damage liver and kidney functions [3]. Mitomycin and adriamycin are cardiotoxic [4,5]. Cytarabine can cause myelosuppression and adverse gastrointestinal reactions [6]. Therefore, researchers are looking towards natural medicines as potential medicines with higher anti-gastric cancer activity and lower toxicity. Currently, polysaccharides are used in biochemistry and medicine due to their remarkable therapeutic effects and low toxicity [7]. Moreover, due to these polysaccharides’ significant anti-gastric cancer activity, they have been used as food additives or medicines [8,9].

In this review, we summarize the anti-gastric mechanisms of polysaccharides, including their ability to induce apoptosis; inhibit proliferation, migration and invasion, and angiogenesis; regulate immune response; and enhance the anticancer effect of anticancer medicines. Sources of these polysaccharides include fungi, algae, *Astragalus membranaceus*, tea, *Caulis Dendrobii,* and other foods and Chinese herbal medicines. In addition, we briefly introduce the extraction, purification, and structural characterization of these polysaccharides. Of these polysaccharides, clinical trials show that the polysaccharide PSK (from the Basidiomycete *Coriolus versicolor*) can significantly improve the therapeutic effect of chemotherapy drugs in gastric cancer patients. However, other polysaccharides are still in the cell experiment or animal experiment phases. We believe that it is necessary to carry out major clinical research to further explore the anti-gastric cancer activity of these polysaccharides. In addition, exploring the relationship between structural characteristics and anti-gastric cancer activity may also be a focus of future research. In short, we hope this review provides guidance for researchers to design, research, and develop polysaccharides.

## 2. Search Strategy

Literature searches were performed using four databases: PubMed, Web of Science (WOS/SCI), Google Scholar, and CNKI (China National Knowledge Infrastructure). Three were English databases (PubMed, Web of Science, and Google Scholar), and one was a Chinese database (CNKI). The keywords used were “polysaccharides”, “plants,” “fungi”, “gastric cancer”, “Anti-gastric cancer”, and “mechanism”. We searched all of the relevant literature published in the last 20 years from 2000 to 2020 that was published in English. The review was completed by searching for bibliographic references and definitions of the topic described above.

## 3. Extraction of Polysaccharides

Polysaccharides are polar macromolecules, which are usually soluble in water but not soluble in organic solvents. Therefore, water is currently used as the extraction medium. As they need to be extracted multiple times using high-temperature distilled water, the water extraction method is simple and easy to operate. In addition, research shows that the use of acid or alkaline solutions instead of distilled water can improve the extraction rate of polysaccharides under certain circumstances [10]. However, although the water extraction method is suitable for almost all polysaccharides, the extraction time can be as long as 2–4 h, which causes the method to take a long time [11]. According to the basic principle of polysaccharide extraction, that is, destroying the cell wall and causing the polysaccharides to enter the solvent, many new extraction techniques have begun to appear.

### 3.1. Ultrasonic Extraction Method

This method uses cavitation to destroy the cell wall to accelerate the dissolution of polysaccharides. Using this method can increase the yield of polysaccharides and decrease the extraction time [12]. By summarizing the research progress on the ultrasonic extraction of Astragalus polysaccharides, Wang et al. found that ultrasonic power had the highest impact on ultrasonic extraction, followed by extraction temperature and extraction time [13]. Therefore, many researchers will screen for the best extraction conditions when using ultrasonic extraction methods [14]. However, it should be noted that exposure to an ultra-sonic environment for a long duration changes the structure of polysaccharides and affects their biological activity [15].

### 3.2. Microwave Extraction Method

When the energy carried by microwaves continues to act on cells, it can increase the intracellular pressure break the cells within a short period of time, and active ingredients such as polysaccharides can flow into the solvent [16]. However, rapid temperature change is very likely to change the molecular weight distribution and the structure of thermally unstable polysaccharides. Research on the microwave extraction of seaweed polysaccharides confirms that this method degrades polysaccharides, resulting in changes in their molecular weight and viscosity [17]. However, microwave extraction is also useful. By changing the microwave power, extraction time, and other factors, researchers can control the degradation rate, sulfate content, viscosity, and molecular weight of seaweed polysaccharides within the required range to obtain the required seaweed polysaccharides [18].

### 3.3. Enzyme-Assisted Extraction

This method destroys the cell wall and intracellular structure through enzymatic hydrolysis to obtain more polysaccharides. The main enzymes that are currently used include Viscozyme, Cellucast, Termamyl, Ultraflo, carragenanase, agarase, amyloglucosidase, xylanase, Kojizyme, Protamex, Neutrase, Flavourzyme, and Alcalase [19,20]. At present, this method is often used in combination with other methods, such as microwave extraction and ultrasonic extraction. Since enzymes are selective to the environment, ensuring enzyme activity is one of the key points to consider when using different enzymes together.

### 3.4. Other Extraction Methods

In addition to the extraction methods mentioned above, there are many new extraction methods that can be applied to polysaccharide extraction, including supercritical CO2 extraction [21], subcritical water extraction [22], ionic liquids extraction [23], and dynamic high-pressure micro-jet technology [24]. In general, there are no absolute advantages and disadvantages between different extraction technologies. Choosing a suitable extraction technology is not only related to the characteristics of the polysaccharides, but is also inseparable from the extraction conditions controlled by the researchers.

## 4. Purification of Polysaccharides

Extracted crude polysaccharides contain impurities such as inorganic salts, proteins, and pigments. Impurities seriously affect the evaluation of the relationship between the structure and biological activity of polysaccharides; as such, they need to be removed. In most cases, ethanol precipitation is the first step in polysaccharide purification, as it can remove low-molecular-weight impurities from polysaccharides [25]. In addition, membrane separation technologies, such as diafiltration and ultrafiltration, are also widely used to remove impurities [26]. Conventional methods of removing protein use Sevag reagent or trichloroacetic acid to denature and precipitate the protein [27]. Methods to remove pigments include the resin method, the activated carbon method, and the hydrogen peroxide oxidation method [15].

To explore the relationship between structure and biological activity, the polysaccharides obtained after removing impurities require further deep purification. At present, the most commonly used method is chromatographic separation. Ion-exchange chromatography is suitable for separating neutral or acidic polysaccharides via gradient salt elution or pH adjustment [28]. For most anti-gastric cancer polysaccharides, dicthylaminoethyl-cellulose anion exchange column chromatography is used for deep purification. Size exclusion chromatography (also known as gel chromatography) is based on the principle of different molecular weights or molecular size for separation [29]. Affinity chromatography uses the adsorption difference between different substances and stationary phases for separation [30].

## 5. Structural Characterization of Polysaccharides

The physical, chemical, and biological properties of polysaccharides mainly depend on the type, ratio, and sequence of monosaccharides; their molecular weight; the configuration of the glycosidic bonds; the types of glycosidic bonds; and the positions of the glycosidic bonds [31]. High-performance gel permeation chromatography can not only be used to determine the homogeneity of polysaccharides, but can also be used to determine the molecular weight [32]. Partial acid hydrolysis, periodic acid oxidation, Smith degradation, high-performance liquid chromatography, gas chromatography, and high-performance thin-layer chromatography are also used to determine the composition of monosaccharides [33]. Nuclear magnetic resonance spectroscopy is used to determine the ratio of monosaccharides and anomeric bonds [34]. Gas chromatography-mass spectrometry is used to determine linkage positions [33]. We summarize the extraction, purification, and characterization steps in Figure 1.

## 6. Polysaccharides with Anti-Gastric Cancer Effect

### 6.1. Protein-Bound Polysaccharide K (PSK)

PSK was obtained from the Basidiomycete *Coriolus Versicolor*, had a molecular weight of 100 kDa, and contained 18–38% protein [35]. It was composed of protein and polysaccharides, had a β-1,4 glucan structure [36], and was made from polysaccharides covalently bonded to peptides through O- or N-glycosidic bonds [37]. PSK has anti-tumor ability and has been used as a nonspecific immunostimulant to treat cancer patients. Additionally, PSK has been combined with various chemotherapeutic agents to act as a novel therapeutic regimen in animal models. In clinical settings, PSK is effective for the treatment of a variety of cancers, including gastric cancer [38]. We summarized the research progress of PSK for gastric cancer treatment in the past 20 years, and research showed that the actions of PSK on anti-gastric cancer include inhibiting the expression of immunosuppressive factors such as TGF-β; activating immune responses, such as promoting the maturation of dendritic cells and correcting Th1/Th2 imbalances; and enhancing the activity of anticancer drugs (Figure 2) [39].

Cell experiments and animal experiments have revealed the mechanism of PSK against gastric cancer (Table 1). The immunomodulatory effect of PSK is related to its anticancer activity. PSK increases the expression of HLA-B27 and β2-microglobulin and increases the expression of HLA-ABC, HLA-A2/A28, and HLA-DR in KATO-3 cells. This suggests that PSK can enhance the immune response against gastric cancer [40]. Research has shown that PSK is a selective TLR2 agonist that modulates immune response by activating dendritic cells and T cells in tumor-bearing neu transgenic mice [41,42]. Research has also shown that PSK induces the activation of natural killer cells as well as of cytotoxicity by enhancing the binding activity of AP-1 and CRE through promoting the expression of PKCδ, PKCε, and ERK3 [43,44]. In addition, PSK can prevent the functional inhibition of dendritic cells caused by the tumor-derived factors of MKN-45P cells, which include increasing phagocytic activity, inducing Th1-type immune regulation, and preventing the apoptosis of dendritic cells [45].

Transforming growth factor-β (TGF-β) is related to immune regulation and cancer development. In a mouse fibrotic tumor model (human peritoneal mesothelial cells and human gastric cancer cell line OCUM-2MD3), PSK inhibited the proliferation and the epithelial-mesenchymal transition by inhibiting the TGF-β-induced overexpression of α-SMA. This suggests that PSK can prevent the migration of gastric cancer to form peritoneal cancer [46]. In addition, PSK inhibits the invasion of MK-1P3 cells by down-regulating the expression of several invasion-related factors, including uPA, MMP-2, and MMP-9 by decreasing TGF-β1 expression [47]. By inhibiting the TGF-β signaling pathway, increasing the expression of E-cadherin, decreasing the expression of vimentin, and inhibiting the epithelial-mesenchymal transition-mediated generation of CD44+/CD24 cells, PSK suppresses the migration and invasion of MKN45 cells [48].

In addition, PSK can improve the therapeutic effect of chemotherapy drugs. PSK enhances docetaxel’s induction of apoptosis and growth inhibition on MK-1 cells. Moreover, PSK can inhibit the invasion of MK-1 cells invasion caused by docetaxel. In a xenograft rat model of MK-1 cells, PSK enhanced docetaxel-mediated tumor suppression by inhibiting the activation of NF-κB and survivin expression [49,50]. In addition, PSK can induce blood mononuclear cells to express IFN-α, inhibiting dihydropyrimidine dehydrogenase expression, and consequently augmenting the antitumor effect of 5-fluorouracil or 5′-deoxy-5-fluorouridine on GCIY and MKN45 cells [36].

On the basis of cell experiments, clinical research also showed that PSK was extremely useful as an adjuvant immunochemotherapy to conventional anticancer drugs [51,52]. PSK prolonged the survival of gastric cancer patients, mainly by regulating immune function, and no serious adverse reactions was reported [53,54,55]. For 8009 gastric cancer curative resection patients, a meta-analysis showed that PSK immunochemotherapy could significantly improve survival rate (P = 0.018) [38]. In 31 advanced gastric cancer patients, the plasma TGF-β levels of the PSK therapy group (*n* = 17) decreased compared to the no PSK group (*n* = 14), thereby antagonizing immune evasion and improving patient prognosis (P = 0.019) [56]. In 310 gastric cancer patients (stage IB: *n* = 19, II: *n* = 65, IIIA: *n* = 58, IIIB: *n* = 28, IV: *n* = 37), the 5-year survival rate increased in the treatment group that received 5-fluorouracil, mitomycin-C, and PSK (84.4%) compared to the treatment group that only received 5-fluorouracil and mitomycin-C (67.6%) (P = 0.019). However, a significant difference was only observed from between stages IIIA and IV (P = 0.019) [57,58,59].

In 918 gastric curative gastrectomy patients (stages II and III), patients with death-1 ligand 1-negative expression in the PSK plus 5-fluorouracil group had longer survival when compared to the 5-fluorouracil-only group (P = 0.033). It is worth noting that the use of PSK was only valid for stages IIIA and IIIB (P = 0.031) and not for stage II or stage IIIC. By examining the patients’ immunological cells, it was observed that the number of natural killer and natural killer T cells increased [60]. For the 14 gastric cancer patients who received tegafur/uracil plus PSK treatment, FACS tests showed that the Th1/Th2 balance shifted to Th1 dominance (the Th1/Th2 ratio from 9.36 to 10.93, P = 0.074), and the DC1/DC2 balance shifted to DC1 dominance (the DC1/DC2 ratio from 1.25 to 1.62, P = 0.090). In addition, ELISA tests showed that IL-10 production significantly decreased (P = 0.015) [61]. In 21 stage III gastric cancer patients, the 3-year overall survival of the PSK plus tegafur/uracil group (62.2%) was better than that of the tegafur/uracil-only group (12.5%) (P = 0.038). FACS tests showed that this was related to decreasing the number of CD57(+) T cells (P = 0.0486) [62]. For 136 stage II or III advanced gastric cancer patients, S-1 (tegafur, gimeracil, and oltipraz) plus PSK treatment reduced the recurrence of patients with advanced non-T4 (P < 0.01) or N0-1 gastric cancer (P = 0.03) [63]. Further testing of the patients’ immunological parameters showed that this was related to decreasing the neutrophil: lymphocyte ratio (P = 0.043) [64], which decreased the expression of the Foxp3 gene (4.26% to 3.11%) and increased natural killer cell activity (27% to 47%) [65], preventing the T-cell apoptosis induced by S-1 by decreasing caspase-3 activity an Bax expression [66].

In the early stages of cancer development, primary tumors are composed of homogeneous HLA class I-positive cancer cells, and then a Darwinian type of T-cell-mediated immune selection results in a tumor solely composed of MHC class I-negative cells, which can be highly diverse and are not necessarily the same as the cells in the primary tumor, and research has shown that cancer cells can avoid macrophage phagocytosis by increasing MHC class I expression [67]. In 349 curative gastric cancer surgery patients, treatment with PSK plus fluoropyrimidine agents significantly improved the 5-year relapse-free survival of patients (MHC class I-positive). Although this treatment method had no effect on MHC class I-negative patients, it had a significant therapeutic effect on other patients (multiple lymph node metastasis and MHC class I-negative). These patients’ 5-year relapse-free survival was also significantly improved [68]. Similarly, in 254 gastric curative resection patients, there was no significant difference in the 8-year overall survival and relapse-free survival between the anti-metabolites group and anti-metabolite plus PSK group. However, the 8-year overall survival rate of the patients with pN3 lymph node metastasis was improved in the anti-metabolites plus PSK group (P = 0.002) [69]. In 349 gastric cancer curative resection patients, the 3-year recurrence-free survival rate was 65% in the PSK plus fluoropyrimidine group and 47% in the fluoropyrimidine-only group. For patients with pN3 lymph node metastasis, the recurrence-free survival rates were 68% in the PSK plus fluoropyrimidine group and 28% in the fluoropyrimidine-only group [70]. In addition, for patients with early tumor recurrence, the therapeutic effect of PSK was obvious. In 254 gastric curative resection patients, the overall survival of patients with early tumor recurrence was better in the fluorouracil plus PSK group than in fluorouracil-only group (P = 0.023) [71]. We summarized the clinical research of PSK as a postoperative adjuvant treatment for patients with gastric cancer in Table 2.

### 6.2. Fungi Polysaccharides

Current research has shown that various fungal polysaccharides (mainly edible mushrooms) have significant anti-gastric cancer effects. For instance, lentinan is a biological response modulator of gastric cancer and demonstrates immunomodulatory activity. It has a (1→3)-β-D-glucan with two (1→6)-β-glucopyranoside branches for every five (1→3)-β-glucopyranoside linear linkages, and its molecular weight is between 304 kDa and 1832 kDa. In Japan, two types of β-glucans, krestin and lentinan, have been licensed as drugs for gastric cancer treatment [72]. β glucans stimulate the immune system through the activation of different immune cells including macrophages, dendritic cells, neutrophils, natural killer (NK), and lymphocytes [73] and are used for cancer therapy in association with cytotoxic chemotherapeutic agents [38]. Additionally, lentinan can activate non-specific cytotoxicity in vivo; enhance helper T cell-mediated cytotoxic T cell activity, NK cell activity, and humoral immune response; induce Th1 polarization; and improve the balance between Th1 and Th2 [72,74].

The anti-gastric cancer effects of fungal polysaccharides are discussed from a variety of different perspectives. (1) Proliferation inhibition: The *Phellinus gilvus* polysaccharide fraction inhibits the proliferation of TMK-1 cells [75]. The polysaccharides HEG-5 and HEP-1 were extracted from *Hericium erinaceus.* HEG-5 inhibited the proliferation of GES-1 cells by blocking cell cycle arrest during the S phase, and HEP-1 inhibited the proliferation of GES-1 cells by blocking the cell cycle during the G0/G1 phase [76,77]. The polysaccharides LSMS-1 and LSMS-2 were also extracted from *Hericium erinaceus* and the average molecular weights were 6842 kDa and 2154 kDa, respectively. MTT assays showed that LSMS-1 and LSMS-2 significantly inhibited the growth of SGC7901 cells [78]. A water-insoluble polysaccharide from *Grifola frondosa* was sulfated to obtain a sulfated polysaccharide S-GAP-P with a molecular weight of 28 kDa and a sulfate content of 16.4%. MTT assays showed that S-GAP-P could inhibit the proliferation of gastric cancer SGC-7901 cells [79]. Yang et al. isolated two polysaccharides FVP-1 and FVP-2 from *Flammulina velutipes*. FVP-1 was composed of glucose, fucose, mannose, and galactose in a ratio of 81.3:3.0:3.6:12.1 and had a molecular weight of 28 kDa, and the FVP-2 was composed of glucose, fucose, xylose, mannose, and galactose in a ratio of 57.9:5.5:9.5:15.1:12.0 and had a molecular weight of 268 kDa. Although MTT assays showed that both FVP-1 and FVP-2 could inhibit the proliferation of gastric cancer BGC-823 cells, the inhibitory strength of FVP-2 was higher than that of FVP-1, with this higher strength being attributed to structural differences [12]. Luo et al. extracted a homogeneous polysaccharide SP1 from *Trametes robiniophila Murr*, which had a molecular weight of 56 kDa [80]. Chen et al. found that SP1 inhibited the proliferation of MGC-803 cells by inhibiting the expression of SOX4, ZEB2, MMP9, Snail, and Slug through promoting SMAD7 expression and inhibiting the activation of the TGF-β/SMAD signaling pathway [81]. The polysaccharide RPS from Rhizopus nigricans was composed of mannose, rhamnose, glucuronic acid, glucose, galactose, and fucose, with a molar ratio of 5.1:1.0:1.6:92.2:1.3:2.3. RPS could inhibit the proliferation of BGC-823 cells by blocking the cell cycle during the G2/M phase [82]. (2) Apoptosis induction: The *Phellinus gilvus* polysaccharide fraction induced the apoptosis of TMK-1 cells and inhibited tumor growth in an orthotopic transplantation mouse model [75]. The polysaccharide POMP2 from *Pleurotus ostreatus* significantly reduced the weight and volume of the tumor in BGC-823 cells xenograft-bearing mice [83]. The polysaccharide HEG-5 induced the apoptosis of GES-1 cells by reducing the expression of Bcl-2 and increasing the expression of caspase-8, caspase-3, p53, Bax, and Bad, and the polysaccharide HEP-1 induced the apoptosis of GES-1 cells by promoting the expression of Bax and caspase-3 and inhibiting the expression of Bcl-2 [76,77]. The protein of the polysaccharide conjugates GFG-3a from *Grifola frondosa* and induced the apoptosis of SGC-7901 cells by increasing the expression of caspase-8, caspase-3, p53, Bax, and Bad, and reducing Bcl-2 and Bcl-xl expression through inhibiting the PI3K/AKT signaling pathway [84]. The sulfated polysaccharide S-GAP-P could induce apoptosis of SGC-7901 cells [79]. The *Ganoderma lucidum* polysaccharide fraction induced the apoptosis of gastric cancer SGC-7901 cells via down-regulating Bcl-2 protein expression and increasing Bax expression [85]. The polysaccharide RPS induced the apoptosis of BGC-823 cells by inducing the production of intracellular ROS and increasing the level of intracellular Ca2+ and the loss of mitochondrial membrane potential, and then activated the expression of caspase-9 and caspase-3 [82]. SP1 induced the apoptosis of MGC-803 cells by inhibiting the expression of SOX4, ZEB2, MMP9, Snail, and Slug through promoting SMAD7 expression and inhibiting the TGF-β/SMAD signaling pathway [81]. (3) Regulating immune activity: The *Ganoderma lucidum* polysaccharide fraction enhanced the immunity and antioxidant activity of Nmethyl-N9-nitro-Nnitrosoguanidine-induced gastric cancer rats by reducing IL-6 and TNF-α levels and increasing levels of IL-2, IL-4, IL-10, SOD, CAT, and GSH-Px [86]. (4) Enhancing the anticancer effect of anticancer medicines: The polysaccharide HEP from *Hericium erinaceus*, which has a molecular weight of 13 kDa, reduced the IC50 value of the doxorubicin-induced apoptosis in SGC7901 cells from 10 μg/mL to 5 μg/mL by increasing doxorubicin-induced ROS production and down-regulating HIF-1α expression [87]. In 39 patients who had been diagnosed with unresectable gastric cancer, the incidence of adverse events during treatment was reduced in comparison to 20 patients who only received S-1 treatment, and quality of life was improved in the 19 patients who were treated with S-1 plus lentinan [88].

### 6.3. Algae Polysaccharides

Current research shows that various algae polysaccharides (mainly brown algae polysaccharides) have significant anti-gastric cancer effects. For instance, laminaran is a water-soluble polysaccharide from the brown alga *Eisenia bicyclis* that has a molecular weight of 5 kDa and a glucan with β-(1→6) side chains linked to a β-(1→3) backbone [89]. Currently, a large number of studies shows the antitumor effects of QSC, with laminaran activating apoptotic cell death, inhibiting the colony formation in cancer cells, and inhibiting the angiogenesis potential of cancer [90]. Regarding the treatment of gastric cancer, oral supplementation with laminaran can attenuate the gross and microscopic signs of gastric dysplasia and can substantially counterbalance the increased induction of the expression of protein markers reminiscent of proliferative and angiogenic processes. Laminaran treatment can effectively overcome A4gnt KO-induced alteration in the gene expression profiles of selected cytokines; thus, β glucan can effectively restrain the progressive development of precancerous lesions and gastric dysplasia in A4gnt KO mice [91].

Its anti-gastric cancer effects are mainly reflected in the following aspects: (1) Apoptosis induction: The polysaccharide fraction from the red alga *Gracilariopsis lemaneiformis* can induce the apoptosis of MKN28 cells by activating the Fas/FasL signaling pathway [92]. The sulfated polysaccharides from the red alga *Porphyra yezoensis* induced AGS cells apoptosis by increasing PARP cleavage and promoting caspase-3 activation through decreasing insulin-like growth factor I receptor (IGF-IR) phosphorylation [93]. Xie et al. extracted a polysaccharide with a molecular weight of 11.68 kDa from a brown alga that induced the apoptosis of MKN45 cells by inducing ROS production, JNK phosphorylation, and the activation of p53, caspase-9, and caspase-3 expression [94,95]. Similarly, the polysaccharide SFPS-B2 extracted from the brown alga *Sargassum fusiforme* induced the apoptosis of SGC-7901 cells by activating the intracellular mitochondrion permeability transition pore, caspase-9, and caspase-3, decreasing the mitochondrial membrane potential and the expression of Bcl-2, up-regulating the expression of Bax, and inducing the release of cytochrome [96]. Cf-PS is a sulfated polysaccharide derived from the brown alga *Capsosiphon fulvescens*. The monosaccharide composition included xylose and mannose at a ratio of 17:3, and the sulfate content was 28.7%. By decreasing the phosphorylation of IGF-IR and the expression of Bcl-2 and increasing the expression of caspase-3, Cf-PS significantly induced the apoptosis of AGS cells [97]. Fucoidan is from the brown alga *Cladosiphon okamuranus*, carries substantial levels of 2-O-α-D-glucuronopyranosyl branches in the linear (1→3)-linked poly-α-fucopyranoside chain, and has a molecular weight of 400 kDa [98]. By downregulating the expression of Bcl-2 and Bcl-xL, inducing the loss of mitochondrial membrane potential and activating the expression of caspase-3, fucoidan induces the apoptosis of AGS cells. Moreover, the conversion of microtubule-associated proteins from LC3-I to LC3-II and the increased accumulation of beclin 1 indicated that fucoidan induces the autophagy of AGS cells [99]. (2) Proliferation activity: By inhibiting the ASK1/p38 signaling pathway, fucoidan inhibits proliferation and DNA synthesis in MKN45 cells [91]. Xie et al. extracted a polysaccharide with a molecular weight of 11.68 kDa from brown alga that inhibited the proliferation of MKN45 cells by blocking the cell cycle during the G2/M phase [94,95]. (3) Angiogenesis inhibition: In A4gnt-KO mice, laminaran inhibited angiogenesis by reducing the expression of IL-10, IL-11, Hgf, Ccl2, IL-1B, Ptgs2, Egf, Fgf7, and Cxcl1 [100]. (4) Immune activity regulation: Glycosyl linkages from the extremely complex sulfated fucoidan SHPPB2 from brown alga *Sargassum henslowianum* improved the immune function of N-methyl-N’-nitro-nitrosoguanidine-induced gastric cancer rats by promoting spleen cell proliferation, increasing the secretion of anti-inflammatory cytokines, and reducing the expression of pro-inflammatory cytokines [101].

### 6.4. Astragalus Membranaceus Polysaccharides

*Astragalus membranaceus* is a widely used Chinese medicine, and the polysaccharides that have been extracted from it have been proven to have anti-gastric cancer effects [102]. (1) Apoptosis induction: The water-soluble polysaccharide APS4 was found to induce the apoptosis of MGC-803 cells by activating the expression of caspase-9/-3 and promoting PARP cleavage through intracellular ROS accumulation, the loss of mitochondrial membrane potential, the release of cytochrome c, the increased expression of Bax, and the reduced expression of Bcl-2 [103]. The water-soluble polysaccharide APS is an α-(1→4)-D-glucan with a single α-D-glucose at the C-6 position every nine residues and a molecular weight of 36 kDa. APS induced the apoptosis of SGC-7901 cells and adriamycin-resistant SGC-7901 cells by enhancing the expression of caspase-3, causing DNA fragmentation and increasing the expression of tumor suppressor genes (SEMA3F, P21WAF1/CIP1, and FBXW7) by activating the MAPK signaling pathway [104]. (2) Proliferation inhibition: The polysaccharide fraction of *Astragalus membranaceus* can inhibit the proliferation of SCG-7901 cells by blocking the cell cycle during the G0/G1 phase [105]. (3) Enhancement of the anticancer effect of anticancer medicines: Research showed that the polysaccharide fraction of *Astragalus membranaceus* enhanced the antitumor effects of Apatinib in AGS cells by inhibiting the proliferation of cancer cells, causing cancer cell apoptosis and autophagy through inhibiting the AKT signaling pathway [106]. (4) Regulation of immune activity: Li et al. isolated a glucan (α-(1→4)-d-glucan with α-(1→6)-linked branches attaching to the O-6 of branch points) that demonstrated an antitumor effect on 1-Methyl-2-nitro-1-nitrosoguanidine-induced gastric cancer mice by promoting spleen lymphocyte proliferation and enhancing natural killer cell activity, raising blood LgA, LgG, and LgM levels and the expression of CD4+ and CD4+/CD8+ [107]. Finally, clinical trials showed that the polysaccharide fraction of *Astragalus membranaceus* reduced gastrointestinal reactions such as nausea, vomiting, and the incidence of leukopenia induced by FOLFOX (composed of oxaliplatin, calcium folinate, and 5-fluorouracil) [108,109].

### 6.5. Tea Polysaccharides

Tea is a popular drink all over the world. Current research shows that tea polysaccharides have an anti-gastric effect [110,111]. (1) Apoptosis induction: Shao et al. found that all three polysaccharides (TSCR, TSCP-1, and TSCP-2) extracted from tea seeds could induce the apoptosis of MKN45 cells, with TSCP-1 demonstrating the strongest anti-tumor activity as well as the highest sulfate content [112]. (2) Proliferation inhibition: Three polysaccharides (TFPS-1, TFPS-2, and TFPS-3) from tea flower were found to inhibit the proliferation of BGC-823 cells. Preliminary characterization showed that TFPS-1 had a sulfate content of 2.63% and a monosaccharide composition that included arabinose, fucose, xylose, mannose, glucose, and galactose, with a molar ratio of 14.84:2.64:12.16:6.87:45.39:18.08. The sulfate contents of TFPS-2 and TFPS-3 were 0.84% and 1.76%, respectively. Both the monosaccharide compositions of TFPS-2 and TFPS-3 contained rhamnose, arabinose, and galactose, and the molar ratios were 11.19:55.16:33.65 and 20.95:53.34:25.71, respectively [113]. (3) Improvement of oxidative stress and inflammation: Three polysaccharides (TF-1, TF-2, and TF-3) from tea were found to protect mice with gastric cancer induced by N-methyl-N’-nitro-nitrosoguanidine from oxidative stress damage and inflammation by increasing the expression of SOD, CAT, and GSH-Px; decreasing the levels of IL-6 and TNF-α; and increasing the levels of immunoglobulin A, immunoglobulin G, immunoglobulin M, IL-2, IL-4, and IL-10. Preliminary characterization showed that the molecular weights of TF-1, TF-2, and TF-3 were 231.58 kDa, 46.27 kDa, and 7.25 kDa, respectively. The monosaccharide composition of TF-1 contained glucose, mannose, and xylose, with a molar ratio of 1:3.2:1.4. The monosaccharide composition of TF-2 contained glucose and xylose, with a molar ratio of 1:1.7. The monosaccharide composition of TF-3 contained glucose, xylose, and arabinose, with a molar ratio of 1:2.5:0.9 [114].

### 6.6. Caulis Dendrobii Polysaccharides

*Caulis Dendrobii* is a precious Chinese herbal medicine, and research has shown that *Caulis Dendrobii* polysaccharides have a therapeutic effect on gastric cancer [115]. (1) Suppression of precancerous lesions: The polysaccharide fraction of *Caulis Dendrobii* inhibited 1-Methyl-2-nitro-1-nitrosoguanidine-induced precancerous lesions in gastric cancer rats by downregulating the gene expression of Wnt2β, Gsk3β, PCNA, CyclinD1, and β-catenin through inhibiting the Wnt/β-catenin signaling pathway [116]. Subsequent research found that it was also related to reducing 8-OHdG levels and activating the NRF2 pathway and the related antioxidant enzymes HO-1 and NQO-1 [117]. (2) Apoptosis induction: The polysaccharide fraction of *Caulis Dendrobii* suppressed the growth of SGC-7901 cell xenografts in nude mice by increasing serum TNF-α and IL-2 levels, upregulating Bax protein expression, and downregulating Bcl-2 protein expression [118]. Liu et al. isolated four polysaccharides, cDHPS, cDHPR, cDHPL, and cDHPF, from *Caulis Dendrobii*. By upregulating the expression of the p53 gene and downregulating the expression of the c-myc gene, cDHPS, cDHPR, cDHPL, and cDHPF could induce the apoptosis of MFC cells [119]. (3) Immune activity regulation: Patients infected with H. pylori have a risk of developing gastric cancer [120]. Studies have shown that in patients with GI cancer receiving anti-PD-1/PD-L1 therapy, regardless of their clinical responses, the gut microbiomes predominantly consisted of Bacteroidetes and Firmicutes, with a 2.5-fold elevation in the *Prevotella*/Bacteroides ratio in patients with favorable outcomes, with a subgroup of responders showing a higher abundance of *Prevotella* [121]. Additionally, animal tests indicated that the *Dendrobium*
*officinale* polysaccharide (DOW-5B) increased the diversity of the gut microbiota in mice, with levels of beneficial microbes such as *Ruminococcus*, *Eubacterium*, *Clostridium*, *Bifidobacterium*, *Parabacteroides*, and *Akkermansia muciniphila* increasing while the levels of harmful Proteobacteria decreased, improving the health of the large intestine as well as the immunity response of the mice [122]. Therefore, we speculate that the *Dendrobium officinale* polysaccharide may inhibit gastric cancer development by improving the gut microbiome, thus improving immune activity.

### 6.7. Other Polysaccharides Extracted from Foods

In addition to the various polysaccharides mentioned above, polysaccharides extracted from some foods have also shown anti-gastric cancer effects. (1) Proliferation inhibition: MTT assays have shown that CFPS-2, CSPS, and LRPs can inhibit the proliferation of SGC-7901 cells. Preliminary characterization showed the average molecular weight of CFPS-2 (from the fresh water clam *Corbicula fluminea*) was about 22 kDa and that the monosaccharide composition contained glucosamine, galactosamine, glucose, galactose, and fucose, with a molar ratio of 0.22:0.15:0.68:0.25:0.86 [123]. CSPS is a selenized polysaccharide obtained from *Capparis spionosa* L. [124]. The α-(1→6)-D-heteroglycans LRPs were extracted from lotus root, and their molecular weights were in the range of 1.33–5.30 kDa. LRPs were mainly composed of Glc-(1→, →6)-Glc-(1→, →6)-Gal-(1→, →4,6)-Gal-(1→ and →3,6)-Glc-(1→ at a molar ratio of 1.00:4.33:0.83:0.13:1.14, and the main monosaccharide composition contained mannose, rhamnose, galacturonic acid, glucose, galactose, and arabinose at a molar ratio of 0.19:0.14:0.17:6.49:1.00:0.16 [125]. In addition, through regulating the expression of cell cycle-associated proteins, cyclins, and CDKs, the polysaccharide fraction of *Lycium barbarum* inhibited the proliferation of MGC-803 and SGC-7901 cells by blocking the cell cycle during the G0/G1 and S phases, respectively [126]. (2) Regulation of immune activity: The α-(1→6)-D-heteroglycans LRPs were extracted from lotus root and were found to significantly enhance the NO production and TNF-α secretion of RAW264.7 macrophages [125]. (3) Migration and invasion suppression: By reducing the expression of MMP2, MMP9, Snail, and vimentin and upregulating the expression of E-cadherin through inhibiting the AKT/PI3K signaling pathway, the polysaccharide fraction of *Lycium barbarum* inhibited the migration and invasion of SGC-7901 cells [127]. (3) Apoptosis induction: Clinical trials showed that after the oral administration of the exocarp polysaccharide fraction of *Ginkgo biloba* in 30 gastric cancer patients, cancer cells appeared to undergo apoptosis and differentiation, and the tumor area was reduced. MTT experiments showed that the exocarp polysaccharide fraction of *Ginkgo biloba* induced the apoptosis of SGC-7901 cells by increasing the expression of the c-fos gene and reducing the expression of the c-myc and Bcl-2 genes [128].

### 6.8. Other Chinese Herbal Medicine Polysaccharides

In addition to the various polysaccharides mentioned above, polysaccharides extracted from certain Chinese herbal medicines have also shown anti-gastric cancer effects. (1) Proliferation inhibition: The polysaccharide AMPS-a (*Abelmoschus manihot* (Linn.) Medicus) has an average molecular weight of 8.8 kDa, and is mainly composed of glucose, mannose, galactose, and fucose, with a molar ratio of 1.00:0.91:2.14:1.09. AMPS-a contains a backbone composed of repeating units of →6)α-D-Galp-(1→6)α-D-Manp-(1→6)α-D-Galp-(1→with β-D-Glcp (1→3)α-Fucp-(1→ branching at the O-3 of mannose and can inhibit the proliferation of MGC-803 and MKN-45 cells [129]. The polysaccharide pMTPS-3 from *Melia toosendan* Sieb. Et Zucc has a molecular weight of 26.1 kDa and contains arabinose, glucose, mannose, and galactose in a molar ratio of 17.3:28.3:41.6:12.6. MTT assays showed that pMTPS-3 could significantly inhibit the growth of BGC-823 cells [130]. The polysaccharide CNP-1-2 from *Clinacanthus nutans* Lindau leaves has a molecular weight of 91.7 kDa and contains rhamnose, arabinose, mannose, glucose, and galactose, with a molar ratio of 1.30:1.00:2.56:4.95:5.09. CNP-1-2 has a backbone consisting of 1,4-linked Glcp, 1,3-linked Glcp, 1,3-linked Manp, 1,4-linked Galp, 1,2,6-linked Galp, and 1,2,6-linked Galp, and it showed a significant growth inhibitory effect on SGC-7901 cells [131]. Polysaccharide P-3 has a molecular weight of 7.8 kDa and was extracted from the leaves of *Magnolia kwangsiensis* Figlar and Noot. It consists of xylose and rhamnose in a ratio of 1:4. MTT assays showed that P-3 could inhibit the growth of SGC7901 cells [132]. MTT assays showed that the polysaccharide fraction of *Radix ranunculi ternati* significantly inhibited the growth and colony formation of BGC823 cells [133]. The α-(1→4)-D-glucan HPS-1 was extracted from the roots of *Hedysarum polybotrys* Hand.-Mazz and was found to have a molecular weight of 94 kDa and a single α-D-glucose at the C-6 position every nine residues. MTT analysis showed that HPS-1 significantly inhibited the proliferation of MGC-803 cells [134]. The polysaccharide CPP was extracted from *Cyclocarya paliurus* (Batal.) Iljinskaja and was found to have a molecular weight of 900 kDa, with a monosaccharide composition containing glucose, rhamnose, arabinose, xylose, mannose, and galactose with molar percentages of 32.7%, 9.33%, 30.6%, 3.48%, 10.4%, and 13.5%. MTT assays have shown that HPS-1 significantly inhibits the proliferation of MGC-803 cells [135]. The polysaccharide PVP from *Prunella vulgaris* is composed of rhamnose, arabinose, xylose, mannose, glucose, and galactose, with a molar ratio of 2.8:28.2:38.5:11.0:3.0:16.5. MTT assays have shown that PVP can inhibit the growth of SGC 901 cells [136]. By blocking the cell cycle at the G1 phase by down-regulating the expression of phosphorylated p-Cdk2T160 and p-Rb and up-regulating the expression of cyclin-dependent kinase2 inhibitor, the polysaccharide fraction of bamboo shavings significantly inhibited the proliferation of six gastric cancer cell lines (AGS, BGC-823, MGC-803, HGC-27, and MKN-45) [137]. (2) Immune activity regulation. By decreasing CD11c+ dendritic cells and CD11b+F4/80+ macrophages in peripheral blood and the spleen, and increasing the levels of CD3+, CD4+, CD8+, and NK1.1+ cells in peripheral blood and the levels of CD19+ and CD11b+ cells in peripheral blood and the spleen, the polysaccharide fraction of bamboo shaving has been shown to inhibit tumor growth and to prolong the survival of forestomach carcinoma tumor-bearing mice [137]. The polysaccharide SMPA extracted from the roots of *Salvia miltiorrhiza* has a molecular weight of 43 kDa and a monosaccharide composition comprising galactose, glucose, rhamnose, mannose, and galacturonic acid in a molar ratio of 2.14:1.42:1.16:2.15:1. By inhibiting the inflammatory response (promoting the production of IL-2, IL-4 and IL-10 and inhibiting the secretion of IL-6 and TNF-α), enhancing the killing activity of natural killer cells and cytotoxic T lymphocytes and the phagocytic function of macrophages, and increasing the levels of total intracellular granzyme-B and IFN-γ, SMPA significantly inhibited tumor growth in N-methyl-N’-nitro-nitrosoguanidine-induced gastric cancer rats [138]. By increasing the peripheral white blood cells count, thymus and spleen indexes, the production of serum cytokines such as IL-2, IL-4, and TNF-α, and promoting splenocytes proliferation, the polysaccharide fraction of *Portulaca oleracea* L. inhibited tumor growth in N-methyl-N’-nitro-N-nitrosoguanidine-induced gastric cancer rats [139]. (3) Migration and invasion suppression: The polysaccharide PGPW1 was extracted from root of Panax ginseng and was found to have a monosaccharide composition that contained glucose, galactose, mannose, and arabinose in a molar ratio of 3.3:1.2:0.5:1.1 [140]. By inhibiting the expression of Twist, AKR1C2, vimentin, and N-cadherin, and upregulating the expression of NF1 and E-cadherin, PGPW1 inhibited the invasion and metastasis of HGC-27 cells [141]. Similarly, the polysaccharide PGP2a was extracted from the root of Panax ginseng. The molecular weight of PGP2a was 32 kDa and the monosaccharide composition contained galactose, arabinose, glucose, and galacturonic acid in a molar ratio of 3.7:1.6:0.5:5.4. By inhibiting the protein expression of Twist and AKR1C2, PGP2a inhibits the migration and invasion of HGC-27 cell [142]. (4) Apoptosis induction: By decreasing the expression of c-myc and bcl-2 and increasing the expression of p53, fas, fas-L, and the cell factor TGF-β1, the polysaccharide fraction of *Acanthopanax giraldii* Harms Var. *Hispidus Hoo* induced the apoptosis of SGC-7901 cells [143]. The polysaccharide WATP was extracted from *Aster tataricus* and was found to have a molecular weight of 91.7 kDa, and a monosaccharide composition that contained galactose, glucose, fucose, rhamnose, arabinose, and mannose in a molar ratio of 2.1:1.3:0.9:0.5:0.3:0.6. By increasing intracellular Ca2+ level and the loss of mitochondrial membrane potential, WATP induced the apoptosis of SGC-7901 cells [144].

## 7. Correlation of Structure and Anti-Gastric-Cancer Activities

Polysaccharides are a class of polymer compounds that are have monosaccharides as their basic constituent units. Structural differences in polysaccharides, including the type and composition ratio of the monosaccharides; the type, position, and number of glycosidic linkages; and the spatial configuration formed by folding, affect the anti-gastric cancer activity of polysaccharides [145,146]. At present, there is not enough research on the structure–function relationship of polysaccharides, so it is difficult to effectively link the structure of polysaccharides with their anti-gastric cancer activity. However, some relationships can be inferred. Shao et al. extracted three kinds of polysaccharides from tea seeds (TSCR, TSCP-1, and TSCP-2) with similar monosaccharide compositions. MTT assays showed that the activity of the three polysaccharides in inducing the apoptosis of MKN45 gastric cancer cells was positively correlated with sulfate content [112]. Similarly, the inhibitory effects of three polysaccharides (TFPS-1, TFPS-2 and TFPS-3) from Camellia sinensis on BGC-823 gastric cancer cells were also positively correlated with sulfate content [113]. In addition, Yang et al. isolated two polysaccharides, FVP-1 and FVP-2, from *Flammulina velutipes*. FVP-1 is composed of glucose, fucose, mannose, and galactose in a ratio of 81.3:3.0:3.6:12.1 with a molecular weight of 28 kDa. The FVP-2 is composed of glucose, fucose, xylose, mannose, and galactose in a ratio of 57.9:5.5:9.5:15.1:12.0 with a molecular weight of 268 kDa. FVP-2 can inhibit the proliferation of BGC-823 gastric cancer cells more strongly than FVP-1. Based on an analysis of the related research, Yang et al. believes that the monosaccharide type, molecular weight, and sulfate and uronic acid content are positively correlated with inhibitory effect of polysaccharides on gastric cancer cells [12]. Liu et al. isolated four polysaccharides: cDHPS, cDHPR, cDHPL, and cDHPF, from *Dendrobium officinale*. cDHPS (259 kDa) was composed of→4)-β-D-Glcp-(1→, →4)-β-D-Manp-(1→, →4)-3-O-acetyl-β-D-Manp-(1→, and had a monosaccharide composition that included mannose and glucose with a molar ratio of 3.04:1.00. cDHPL (209 kDa) was composed of→4)-β-D-Glcp-(1→, →4)-β-D-Manp-(1→, →4)-3-O-acetyl-β-D-Manp-(1→, →3,6)-β-D-Manp-(1→ and terminal α-D-Galp and had a monosaccharide composition that included mannose, glucose, and galactose with a molar ratio of 19.15:1.32:1.00. cDHPF (478 kDa) was composed of→4)-β-D-Glcp-(1→, →4)-β-D-Manp-(1→, →3,6)-β-D-Manp-(1→ and terminal α-D-Galp and had a monosaccharide composition that included mannose, glucose, and galactose with a molar ratio of 9.68:3.26:1.00. cDHPR (14.1 kDa) was composed of →3,5)-α-L-Araf-(1→, →4)-β-D-Glcp-(1→, →4)-β-D-Manp-(1→, →4,6)-β-D-Manp-(1→, →6)-α-D-Galp-(1→ and terminal β-L-Araf and had a monosaccharide composition that included mannose, glucose, galactose, and arabinose at a molar ratio of 2.38:1.00:8.49:5.23. Among these four polysaccharides, cDHPS demonstrated the strongest activity in proliferation inhibition and in inducing apoptosis, followed by cDHPL, cDHPF, and cDHPR. By comparing the differences between the structures, it can be seen that glycosidic linkage types and sequences also greatly affect the activity of polysaccharides against gastric cancer. Moreover, different from the previous research conclusions of Yang et al., although cDHPS had the fewest types of monosaccharide, and a molecular weight that was not the largest, it had the highest anti-gastric cancer activity. In addition, although types of monosaccharide composition were the same and the ratios of the monosaccharide composition were close, cDHPF had a larger molecular weight than cDHPF and cDHPL, which meant that the influence caused by the glycosidic linkage types and sequences made the anti-cancer activity of cDHPF lower than that of cDHPL [119]. Since the 1930s, studies in gastric cancer patients, melanoma, and leukemia treated with BCG showed disease remission or a non-relapsing disease [147]. In addition, what stands out is that PSK, lentinan, and laminaran, all derived from different biological sources, are beta glucans having 1-6 and 1-3 linkages, and many of the other anti-cancer polysaccharides described in the review are alpha glucans. This suggests that one possible origin of the immunomodulatory activity of glucans may be defense against mycobacteria, since the surface integument of *M. bovis* BCG consists largely of an alpha glucan that may account for most of BCG’s anti-cancer activity, possibly as a T-cell stimulant [148,149,150]. In short, the influence of the polysaccharide structure on anti-gastric cancer activity needs to be the subject of major research in the future. This is of great significance for modifying polysaccharides to improve their anti-gastric cancer activity.

## 8. Impacting on the Signaling Pathways and Immune System

By reviewing the above literature, it can be seen that there are more than 60 polysaccharides (including single polysaccharide components and polysaccharide fractions) that have the potential to be used in the treatment of gastric cancer. Despite the large number of studies, most research shows that these polysaccharides have anti-gastric cancer effects that regulate different signal pathways, including the inhibition of the TGF-β, AKT/PI3K, IGF/IR, and Wnt/β-catenin signaling pathways, and activate the MAPK and Fas/FasL signaling pathways (Figure 3). The anti-gastric cancer signaling mechanisms of polysaccharides can be summarized as follows: They inactivate Smad2 signaling and inhibit the expression of SOX4, ZEB2, MMP9, Snail, and Slug by inhibiting the activation of the TGF-β signaling pathway [81]. Additionally, they also reduce the expression of MMP-2, MMP-9, Snail, and vimentin and upregulate the expression of E-cadherin through inhibiting the AKT/PI3K signaling pathway and inducing the apoptosis of SGC-7901 cells by increasing the expression of caspase-8, caspase-3, p53, Bax, and Bad and by reducing Bcl-2 and Bcl-xl expression through inhibiting the PI3K/AKT signaling pathway [84]. Polysaccharides also induce apoptosis in AGS gastric cancer cells via an IGF-IR-mediated PI3K/Akt pathway or by increasing PARP cleavage and Bcl-2 and by promoting caspase-3 and Cf-PS activation by decreasing the phosphorylation of IGF-IR; they also induce the apoptosis of MKN28 cells by activating the Fas/FasL signaling pathway and induce the apoptosis of SGC-7901 cells by decreasing the expression of c-myc and bcl-2 and increasing the expression of p53, fas, fas-L, and the cell factor TGF-β1 [97]. In addition, polysaccharides also inhibit 1-methyl-2-nitro-1-nitrosoguanidine-induced precancerous lesions in gastric cancer rats by downregulating the gene expression of Wnt2β, Gsk3β, PCNA, CyclinD1, and β-catenin by inhibiting the Wnt/β-catenin signaling pathway [116]. They also regulate the expression of cell cycle-associated proteins, cyclins, and CDKs by inhibiting the proliferation of MGC-803 and SGC-7901 cells by blocking the cell cycle during the G0/G1 and S phases, respectively; downregulating the expression of phosphorylated p-Cdk2T160 and p-Rb [126] and upregulating the expression of the CDK2 inhibitor by blocking the cell cycle at the G1 phase [137]; and inhibiting the proliferation of BGC-823 cells and MKN45 cells by blocking the cell cycle at the G2/M phase [82]. Moreover, polysaccharides induce the apoptosis of SGC-7901 cells and adriamycin-resistant SGC-7901 cells by enhancing the expression of caspase-3, causing DNA fragmentation and increasing the expression of tumor suppressor genes (SEMA3F, P21WAF1/CIP1, and FBXW7) by activating the MAPK signaling pathway [104]. They can also induce the apoptosis of GES-1 cells by reducing the expression of Bcl-2 and increasing the expression of caspase-8, caspase-3, p53, Bax, and Bad, and the polysaccharide HEP-1, as well as by inhibiting the expression of Bcl-2 [76,77]. Polysaccharides can also inhibit the proliferation and DNA synthesis in MKN45 cells by inhibiting the ASK1-p38 signaling pathway and inducing ROS production and JNK phosphorylation and by activating the expression of p53, caspase-9, and caspase-3 [94,95].

Through above effects, polysaccharides can induce apoptosis of gastric cancer cells and inhibit the proliferation, migration, and invasion of gastric cancer. However, although research shows that the anti-gastric cancer effects of polysaccharides involve multiple mechanisms from a holistic perspective, we speculated that there are still some important signaling pathways and related proteins and nucleic acids that have not been revealed. Moreover, when focusing on each specific polysaccharide, it can be found that the anti-gastric cancer research on some polysaccharides is superficial, that is, it only shows the inhibitory effect on the growth of gastric cancer cells and does not reveal more mechanisms of action. In addition, some research shows that polysaccharides can kill gastric cancer cells by activating the immune system (Figure 4). However, in addition to PSK, only the polysaccharide fractions of *Ganoderma lucidum*, bamboo shavings, SHPPB2, LRPs, and a glucan from *Astragalus membranaceus* show immunomodulatory activity.

## 9. Conclusions

At present, researchers are gradually paying more attention to the anti-gastric cancer effects of polysaccharides, and these polysaccharides also show great developmental potential. We summarized the mechanism of polysaccharides against gastric cancer (Table 3). Although the polysaccharide PSK has achieved significant clinical therapeutic effects, pharmacological research on other polysaccharides is still in the cell experiment or animal experiment stages. Moreover, the molecular mechanisms obtained from cell or animal experiments show a weak correlation with the actual role of polysaccharides in the human body. Therefore, we believe that more clinical trials are required in the future to determine the therapeutic effects of these polysaccharides. In addition, because the molecular structures of these polysaccharides are quite complex, the significant degradation caused by the first-pass effect needs to be considered. Therefore, when conducting clinical research in the future, the data obtained from cell experiments and animal experiments need to be used with caution. In addition, we recommend using omics technologies and bioinformatics to further explore the therapeutic mechanism of polysaccharides. In addition, the relationship between the biological activity of polysaccharides and their chemical structure needs to be explored. The biological characteristics of polysaccharides are closely related to their chemical structures, the main components of which are their monosaccharide composition, molecular weight, and type and location of their glycosidic bonds. The current literature research has initially shown that the structure has a non-negligible effect on anti-gastric cancer activity [12,112,119]. Finally, we speculate that polysaccharides may inhibit gastric cancer development by improving the gut microbiome, thus improving immune activity. Thus, there is much room to search for the intrinsic link between polysaccharides and intestinal microbiota for the treatment of gastric cancer.

In short, we have detailed the anti-gastric cancer effects of plant and fungal polysaccharides, and hope that review is helpful for our researchers to design, research, and develop polysaccharides for the treatment of gastric cancer.

## Figures and Tables

**Figure 1 molecules-27-05828-f001:**
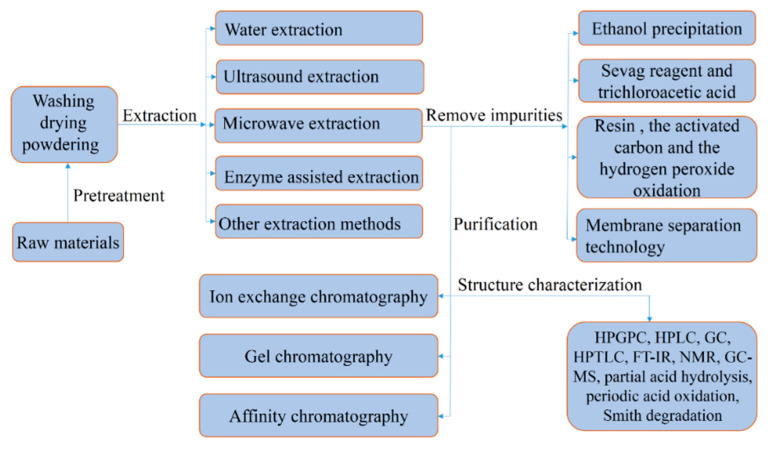
Schematic representation of the extraction, purification, and characterization of polysaccharides against gastric cancer.

**Figure 2 molecules-27-05828-f002:**
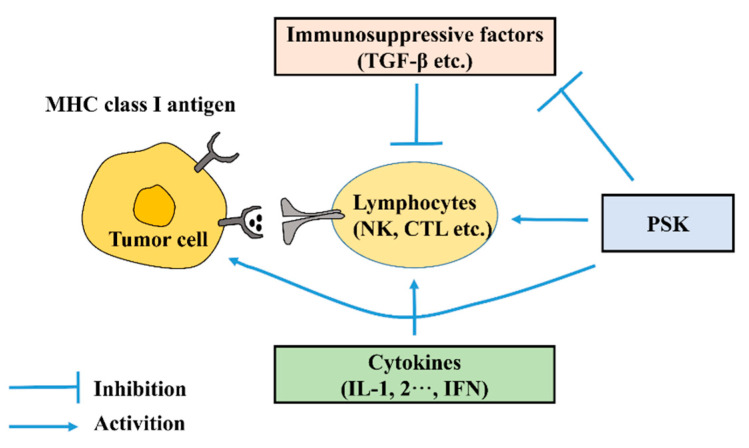
Tumor microenvironment and actions of PSK.

**Figure 3 molecules-27-05828-f003:**
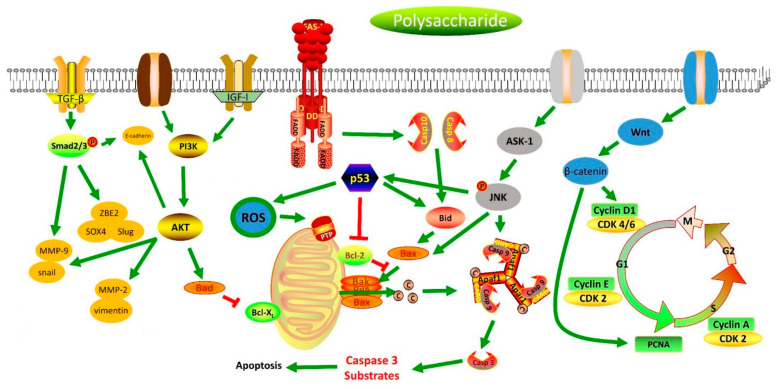
Scheme summarizing the polysaccharides’ signaling mechanisms of anti-gastric-cancer.

**Figure 4 molecules-27-05828-f004:**
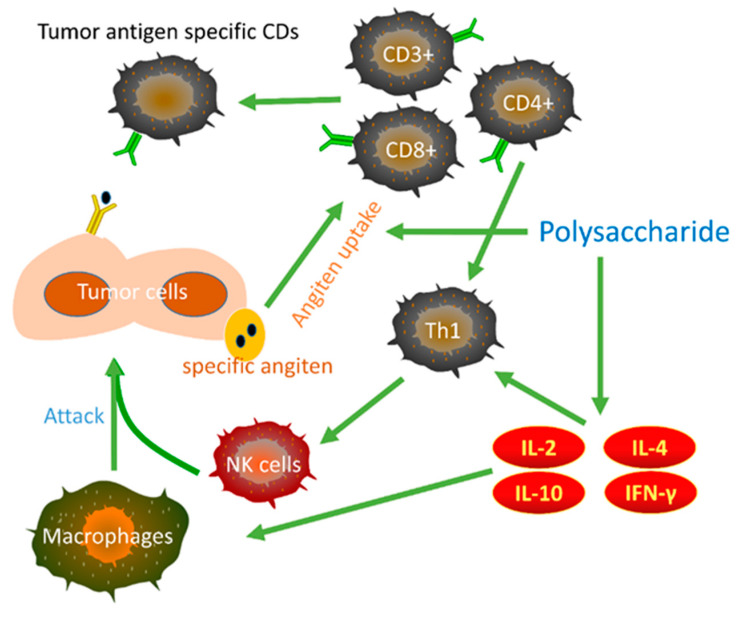
Schematic representation of the antitumor immune responses and actions of polysaccharides.

**Table 1 molecules-27-05828-t001:** Direct actions of PSK on gastric cancer cells.

Detailed Mechanism of Action	Cell Lines/Model	Refs.
Increase the expression of HLA-ABC, HLA-A2/A28, HLA-DR, HLA-B27, and β2-microglobulin.	KATO-3 cells	[40]
Prevent functional inhibition of dendritic cells caused by tumor-derived factors	MKN-45P cells	[45]
Inhibit the TGF-β-induced overexpression of α-SMA	Mouse fibrotic tumor model	[46]
Down-regulate TGF-β1, uPA, MMP-2, and MMP-9 expression	MK-1P3 cells	[47]
Decrease the expression of several TGF-β pathway target genes	MKN45 cells	[48]
Enhance effect of docetaxel’s induction of apoptosis and growth inhibition and reduce docetaxel-induced invasion	MK-1 cells and xenograft mice	[49,50]
Induce blood mononuclear cells to express IFN-α	MKN45 cells	[36]

**Table 2 molecules-27-05828-t002:** Postoperative adjuvant chemotherapy with PSK for resected gastric cancer.

Stage	No. of Patients	Treatment	The Effect of Adding PSK	Refs.
IIIA/IIIB	560	5-FU or tegafur/uraci	Improve prolonged patient survival (P = 0.031), especially for the PD-L1 (-) subgroup (P = 0.033)	[59]
	380	PSK and 5-FU or tegafur/uraci		
Unknown	14	PSK and tegafur/uraci	Th1 and DC1 were dominance, and IL-10 production decreased from 559.36 ± 147.08 pg/mL to 422.14 ± 219.29 pg/mL(P = 0.015)	[60]
	6 (Healthy person)	PSK and tegafur/uraci		
III/IV	14	No treatment	Inhibite plasma TGF-β (from 21.6 to 4.5 ng/mL, on average) and thereby antagonize immune evasion	[55]
	17	PSK		
IB, II, IIIA, IIIB, IV	207	5-FU, MMC and PSK	Improve the 5-year survival rate	[56,57,58]
	103	5-FU and MMC		
III	11	UFT	Decrease CD57(+) T cells significantly (P = 0.0486)	[61]
	10	UFT and PSK		
II	138	Fluorouracil	Increase overall survival for patients with early tumor recurrence (P = 0.023) and median overall survival for patients with pN3 lymph node metastasis (P = 0.032)	[70]
	115	Fluorouracil and PSK		
II and III	61	S-1 adjuvant chemotherapy	Prolong the treatment cycles (S-1 plus PSK was significantly higher than that of S-1 alone (P < 0.01); (P = 0.041) to reduce the recurrence rate	[62,71]
	75	S-1 adjuvant chemotherapy and PSK		
II and III	225	Fluoropyrimidine agents	Reduce tumor recurrence rate (the negative patients of the 3-year recurrence-free survival was 62%) through up-regulating the expression of MHC class I	[67]
	124	Fluoropyrimidine agents and PSK		
II	138	Antimetabolites	Increase the overall survival for patients with pN3 and early tumor recurrence	[68]
	115	Antimetabolites and PSK		

**Table 3 molecules-27-05828-t003:** Anti-gastric cancer effect of polysaccharides.

Sources	Name	Monosaccharides Composition	Mw/kDa	Effects	Refs.
Tea	TF-1	Glucose:mannose:xylose = 1:3.2:1.4	231.5	Protect gastric cancer mice from oxidative stress damage and regulate immune response	[114]
	TF-2	Glucose:xylose = 1:1.7	46.2		
	TF-3	Glucose:xylose:arabinose = 1:2.5:0.9	7.2		
	TFPS-1	Arabinose:fucose:xylose:mannose:glucose:galactose = 14.84:2.64:12.16:6.87:45.39:18.08	-	Inhibit the proliferation of human gastric cancer BGC-823 cells	[113]
	TFPS-2	Rhamnose:arabinose:galactose = 11.19:55.16:33.65	-		
	TFPS-3	Rhamnose:arabinose:galactose =20.95:53.34:25.71	-		
	TSCR, TSCP-1and TSCP-2	-	-	Induce apoptosis of MKN45 gastric cancer cells	[112]
*Coriolus Versicolor* mushroom	PSK	Glucose:mannose:xylose:galactose:fucose = 56:11.6:3.6:2.0:1.8	100	Inhibit the expression of immunosuppressive factors such as TGF-β, activate the immune response such as promoting maturation of dendritic cells, and correct imbalance of Th1/Th2, and enhance the activity of anticancer drugs	[39,151]
*Hericium erinaceus*	HEG-5 and HEP-1	-	-	Cause cell cycle arrest and induce expression of apoptotic proteins	[76,77]
	HEP	1,3-branched-β-1,6-glucan with a triple helix conformation	13	Promote the anticancer activity of doxorubicin	[87]
Rhizopus nigricans	RPS	Mannose:rhamnose:glucuronic acid:glucose:galactose:fucose = 5.1:1.0:1.6:92.2:1.3:2.3	-	Activate caspase-9 and caspase-3 expression and cause cell cycle arrest	[82]
*Trametes robiniophila Murr*	SP1	a backbone consisting of 1,4-linked-β-D-Galp and 1,3,6-linked-β-D-Galp residues, which was terminated with 1-linked-α-D-Glcp and 1-linked-α-L-Araf terminal at O-3 position of 1,3,6-linked-β-D-Galp unit along the main chain in the ratio of 1.1:2.0:1.1:1.1	56	Inhibit the proliferation, migration, and invasion of gastric cancer MGC-803 cells and induce apoptosis	[81]
*Grifola frondosa*	GFG-3a	-	-	Cause cell cycle arrest and induce expression of apoptotic proteins	[84]
	S-GAP-P	-	28	Inhibit the proliferation of SGC-7901 cells and induce apoptosis	[79]
*Flammulina velutipes*	FVP-1	Glucose:fucose:mannose:galactose = 81.3:3.0:3.6:12.1	28	- Inhibit the proliferation of gastric cancer BGC-823 cells	[12]
	FVP-2	Glucose:fucose:xylose:mannose:galactose = 57.9:5.5:9.5:15.1:12.0	268		
*Ganoderma lucidum*	Polysaccharides fraction	-	-	Inhibit the proliferation and induce apoptosis on gastric cancer SGC-7901 cells	[85]
	Polysaccharides fraction	-	-	Enhance immunity and antioxidant activity of gastric cancer rats	[86]
*Pleurotus ostreatus*	POMP2	-	29	Reduce the weight and volume of the tumor in BGC-823 cells xenograft-bearing mice	[83]
*Lentinula edodes*	Lentinan	(1→3)-β-D-glucan having two (1→6)-β-glucopyranoside branches for every five (1→3)-β-glucopyranoside linear linkages	304–1832	Improve patients’ side effects caused by S-1 treatment	[88]
	LSMS-1	-	6842	Inhibit the growth of SGC7901 gastric cancer cells	[78]
	LSMS-2	-	2154		
*Phellinus gilvus* Algae	Polysaccharides fraction	-	-	Inhibit the growth of tumor tissue in gastric cancer mice	[75]
	-	-	11.68	Inhibit proliferation and induce apoptosis of gastric cancer MKN45 cells	[94,95]
	Algae p Polysaccharides fraction	-	-	Activate the Fas/FasL signaling pathway	[92]
	Laminaran	a glucan with β-(1→6) side chains linked to a β-(1→3) backbone with relatively few branch points, and the ratio of the β-(1→3) and β-(1→6) linkages was 1.5:1	5	Improve gastric dysplasia development and inhibit the increased induction in cell proliferation and angiogenesis in A4gnt KO mice	[89,100]
	SHPPB2	-	-	Promote the proliferation of spleen cells and increase the secretion of anti-inflammatory cytokines in gastric cancer mice	[101]
	Cf-PS	xylose:mannose = 17:3	-	Inhibit the proliferation and apoptosis of AGS gastric cancer cells	[97]
	fucoidan	2-O-α-D-glucuronopyranosyl branches in the linear (1→3)-linked poly-α-fucopyranoside chain	400	Inhibit ASK1/p38 signaling pathway in MKN45 cells and induce apoptosis and autophagy of AGS gastric cells	[91,98,99]
*Astragalus membranaceus*	Polysaccharides fraction	-	-	Inhibit proliferation and induce apoptosis of gastric cancer AGS and SCG-7901 cells	[105,106]
	Polysaccharides fraction	-	-	Reduce gastrointestinal reactions induced by FOLFOX	[108,109]
	APS4	-	-	Induce apoptosis of MGC-803 cells by increasing the expression of pro-apoptotic proteins	[103]
	-	α-(1→4)-d-glucan with α-(1→6)-linked branches attached to the O-6 of branch points	-	Promote proliferation of spleen cells and expression of immune factors in gastric cancer mice	[107]
	APS	α-(1→4)-D-glucan, with a single α-D-glucose at the C-6 position every nine residue	36	Induce apoptosis of SGC-7901 and adriamycin-resistant SGC-7901 cells by activating MAPK signaling pathway	[104]
*Caulis Dendrobii*	Polysaccharides fraction	-	-	Inhibit 1-Methyl-2-nitro-1-nitrosoguanidine induced precancerous lesions of gastric cancer in rats	[116,117]
	Polysaccharides fraction	-	-	Suppress the growth of SGC-7901 cell xenografts in nude mice	[118]
	cDHPS	→4)-β-D-Glcp-(1→, →4)-β-D-Manp-(1→, →4)-3-O-acetyl-β-D-Manp-(1→, and mannose:glucose = 3.04:1.00	259	By up-regulating the expression of the p53 gene and downregulating the expression of c-myc gene, cDHPS, cDHPR, cDHPL, and cDHPF could induce apoptosis of MFC cells	[119]
	cDHPR	→3,5)-α-L-Araf-(1→, →4)-β-D-Glcp-(1→, →4)-β-D-Manp-(1→, →4,6)-β-D-Manp-(1→, →6)-α-D-Galp-(1→ and terminal β-L-Araf, and mannose:glucose:galactose:arabinose = 2.38:1.00:8.49:5.23	14.1		
	cDHPL	→4)-β-D-Glcp-(1→, →4)-β-D-Manp-(1→, →4)-3-O-acetyl-β-D-Manp-(1→, →3,6)-β-D-Manp-(1→ and terminal α-D-Galp, and mannose:glucose:galactose = 19.15:1.32:1.00	209		
	cDHPF	→4)-β-D-Glcp-(1→, →4)-β-D-Manp-(1→, →3,6)-β-D-Manp-(1→ and terminal α-D-Galp, and mannose:glucose:galactose = 9.68:3.26:1.00	478		
clam *Corbicula fluminea*	CFPS-2	Glucosamine:galactosamine:glucose:galactose:fucose = 0.22:0.15:0.68:0.25:0.86	22	Inhibit the growth inhibition of gastric cancer SGC7901 cells	[123]
*Lycium barbarum*	polysaccharide fraction	-	-	Inhibit the proliferation of gastric cancer MGC-803 and SGC-7901 cells and suppress migration and invasion of SGC-7901 cells	[126,127]
lotus seeds	LRPs	Glc-(1→, →6)-Glc-(1→, →6)-Gal-(1→, →4,6)-Gal-(1→ and →3,6)-Glc-(1→ at a molar ratio of 1.00:4.33:0.83:0.13:1.14 and mannose:rhamnose:galactose:glucose:galactose:arabinose at = 0.19:0.14:0.17:6.49:1.00:0.16	1.33–5.30	Inhibit proliferation of gastric cancer SGC7901 cells	[125]
*Ginkgo biloba* exocarp	Polysaccharides fraction	-	-	Induce apoptosis and differentiation in 30 gastric cancer patients	[128]
*Abelmoschus manihot* (Linn.) Medicus	AMPS-a	contained a backbone composed of repeating units of →6)α-D-Galp-(1→6)α-D-Manp-(1→6)α-D-Galp-(1→ with β-D-Glcp (1→3)α-Fucp-(1→ branching at O-3 of mannose, and glucose:mannose:galactose:fucose = 1.00:0.91:2.14:1.09	8.8	Inhibit the proliferation of gastric cancer MGC-803, MKN-45 cells	[129]
Bamboo shaving	Polysaccharides fraction	-	-	Inhibit the proliferation of 6 gastric cancer cell lines and regulate the immune activity of gastric cancer mice	[137]
*Melia toosendan* Sieb. Et Zucc	pMTPS-3	Arabinose:glucose:mannose:galactose = 17.3:28.3:41.6:12.6	26.1	Inhibit the growth of gastric cancer BGC-823 cells	[130]
*Acanthopanax giraldii* Harms Var. *Hispidus Hoo*	Polysaccharides fraction	-	-	Inhibit the proliferation, the colony forming ability and function of SGC-7901 gastric cancer cells	[143]
*Salvia miltiorrhiza*	SMPA	galactose:glucose:rhamnose:mannose:galacturonic acid = 2.14:1.42:1.16:2.15:1	43	Suppress inflammation and activate immune response in Gastric cancer mice	[138]
*Clinacanthus nutans* Lindau	CNP-1-2	a backbone consisting of 1,4-linked Glcp, 1,3-linked Glcp, 1,3-linked Manp, 1,4-linked Galp, 1,2,6-linked Galp and 1,2,6-linked Galp, and rhamnose:arabinose:mannose:glucose:galactose = 1.30:1.00:2.56:4.95:5.09	91.7	Inhibit the growth inhibition of gastric cancer cell SGC-7901	[131]
*Magnolia kwangsiensis* Figlar & Noot	P-3	Xylose:rhamnose = 1:4	7.8	Inhibit the growth of SGC7901 cells	[132]
*Radix ranunculi ternati*	Polysaccharides fraction	-	-	Inhibit growth and colony formation of gastric cancer BGC823 cells	[133]
*Portulaca oleracea* L.	Polysaccharides fraction	-	-	Increase peripheral white blood cells count, thymus and spleen indexes and production of serum cytokines	[139]
*Hedysarum polybotrys* Hand.-Mazz	HPS-1	α-D-glucose at the C-6 position every nine residues, on average	94	Inhibit the proliferation of gastric cancer MGC-803 cells	[134]
*Cyclocarya paliurus* (Batal.) Iljinskaja	CPP	Glucose, rhamnose, arabinose, xylose, mannose and galactose with molar percentages of 32.7%, 9.33%, 30.6%, 3.48%, 10.4%, and 13.5%	900	Inhibit the proliferation of gastric cancer MGC-803 cells	[135]
Panax ginseng	PGPW1	Glucose:galactose:mannose:arabinose = 3.3:1.2:0.5:1.1	350	Inhibit the invasion and metastasis of gastric cancer HGC-27 cells	[140,141]
	PGP2a	Galactose:arabinose:glucose:galacturonic acid = 3.7:1.6:0.5:5.4	32	Inhibit the proliferation, migration and invasion of HGC-27 cells	[142]
*Prunella vulgaris*	PVP	Rhamnose:arabinose:xylose:mannose:glucose:galactose = 2.8:28.2:38.5:11.0:3.0:16.5	-	Inhibit the growth of gastric cancer SGC 901 cells	[136]
Aster tataricus	WATP	Galactose:glucose:fucose:rhamnose:arabinose:mannose = 2.1:1.3:0.9:0.5:0.3:0.6	63	Induces apoptosis of human gastric cancer SGC-7901 cells	[144]

## Data Availability

Not applicable.

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
