# Peer review of "Research Progress on the Mechanisms of Polysaccharides against Gastric Cancer"

_molecules, 2022, doi:10.3390/molecules27185828_

Round 1
Reviewer 1 Report
The authors have attempted to summarize a sizable literature regarding the effects of polysaccharides derived from natural sources on gastric cancer and to infer structure-function relationships from the diverse data, an effort that was largely unsuccessful. What has resulted is a catalogue of disparate information about sources, saccharide compositions, linkages, transduction pathways affected, endpoints, and host/tumor processes stimulated or inhibited, having no instructive or useful effect. In the end, the authors fall back on the easy platitude of "much more work/research is needed." In fact, detailed research on the effect of polysaccharides on cancer goes back more than 50 years. The authors should have selected a number of well-characterized polysaccharides, such as the PSK, lentinan and laminaran that they mention and summarize their effects on cancer in general, including the anti-cancer mechanisms they affect, the polysaccharide structures/composition that impact these mechanisms, and what general principles are responsible for the connection. For instance, many immune mechanisms stimulated by polysaccharides are due to evolutionarily adaptive recognition of pathologic microbes and the immune mechanisms effective against them. Direct effects on cancer cell metabolism, including cell cycle arrest, may have to do with defense against parasites or transformational changes in cell surface glycans. The authors could then examine the extant information about polysaccharides activity against gastric cancer that they present for unifying explanations of their observed effects and directions for further study. It is not the reader's responsibility to make sense of these data, but the authors'.
Specific comments) On page 1, I think the authors meant to say that gastric cancer is the third leading cause of cancer death. On page 6, the passage regarding MHC Class I negative cells is not clear. Table 2 is not very informative. Quantitative data should be stated regarding the effect of adding PSK (e.g., How much was the survival rate improved? How much was IL-10 production decreased?). For much of the information summarized in the following sections, one wonders how cancer-specific these effects are and why. What is the implication of blocking E-cadherin, one of the major direct effects of anti-cancer polysaccharides summarized in Figure 3? Does it impair uptake of nutrients into cancer cells? Why isn't PSK included in Table 3?
Author Response
Dear reviewer,
Please find attached response.

Reviewer 2 Report
In my opinion the review raises a very important and interesting problem.
A large amount of literature has been collected and discussed in the manuscript.
The authors should provide some information which databases and what kyewords were used for the searching.
Latin names of plant and mushroom species should be in italics.
English language should be checked by native speaker and corrected to more formal and scientific.
I would consider changing the title as extraction and purification are not covered in detail but only broadly mentioned.
Author Response
Dear reviewers,
Re:Manuscript ID:molecules-1883212 and Title:Research progress on the against gastric cancer mechanism of polysaccharides.
Thank you very much for your comments and professional advice. These opinions help to improve academic rigor of our article. Based on your suggestion and request, we have made corrected modifications on the revised manuscript. Meanwhile, the manuscript had be reviewed and edited by MDPI English Editing and our English editing ID is English-49563. We hope that our work can be improved again. Furthermore, we would like to show the details as follows.
Reviewer·2#
1.Introduction “The authors should provide some information which databases and what kyewords were used for the searching.”
The auther’ s answer:We agree and have provided these information in manuscript on page 2. Details as follows: “Search strategy: Literature searches were performed using four databases: PubMed, Web of Science (WOS/SCI), Google Scholar, and CNKI (China National Knowledge Infrastructure). Three were English databases (PubMed, Web of Science, and Google Scholar), and one was a Chinese database (CNKI). The keywords used were “polysaccharides”, “plants,” “fungi”, “gastric cancer”, “Anti-gastric cancer”, and “mechanism”. We searched all of the relevant literature published in the last 20 years, from 2000 to 2020, and that was published in English. The review was completed by searching for bibliographic references and definitions of the topic described above.”
2. Introduction“Latin names of plant and mushroom species should be in italics.”
The auther’ s answer:These have been changed to italics in manuscript.
3. Introduction“English language should be checked by native speaker and corrected to more formal and scientific.”
The auther’ s answer:Our manuscript has be reviewed and edited by MDPI English Editing and we have offered English-Editing-Certificate-49563. (Please see the attachment for details) Then we hope that our manuscript meets your requirements.
4. Introduction“I would consider changing the title as extraction and purification are not covered in detail but only broadly mentioned.”
The auther’ s answer:We agree and have changed the title which is “ Research progress on the against gastric cancer mechanism of polysaccharides.”
We would like to thank you again for taking the time to review our manuscript.We look forward to hearing from you.
Sincerely yours,
School of Food and Biological Engineering, Xihua University
Liping Chen lpingchen2022@163.com (L.C.)
Ling Li 0120030074@mail.xhu.edu.cn (L.L.)
Tel. / Fax: +86 15184471605.
30 Aug., 2022

Round 2
Reviewer 1 Report
I am surprised how quickly the authors were able to make the major changes that I requested. The manuscript is much improved and now actually provides important insights into the nature of polysaccharide anticancer mechanisms. What stands out is that PSK, lentinan and laminaran, all derived from different biological sources, are beta glucans having 1-6 and 1-3 linkages. Many of the other anti-cancer polysaccharides described in the review are alpha glucans. This suggests that one possible origin of the immunomodulatory activity of glucans may be defense against mycobacteria, since the surface integument of M. bovis BCG consists largely of an alpha glucan that may account for most of BCG's anti-cancer activity, possibly as a T-cell stimulant (1-3). The authors should re-examine the revised manuscript for English errors, since I noticed several misspellings and syntactical errors in the revised text.
References
1. Wang, R., Klegerman, M.E., Marsden, I., Sinnott, M., and Groves, M.J.: An Antineoplastic Glycan Isolated from Mycobacterium bovis BCG Vaccine. Biochem. J. 311:867-72 (1995).
2. Klegerman, M.E., Devadoss, P.O., Garrido, J.L., Reyes, H.R., and Groves, M.J.: Chemical and Ultrastructural Investigations of Mycobacterium bovis BCG: Implications for the Molecular Structure of the Mycobacterial Cell Envelope. FEMS Immunol. Med. Microbiol. 15:213-22 (1996).
3. Garrido, J.L., Klegerman, M.E., Reyes, H.R., and Groves, M.J.: Antineoplastic Activity of BCG: Location of Antineoplastic Glycans in the Cellular Integument of Mycobacterium bovis, BCG Vaccine, Connaught Substrain. Cytobios 90:47-65 (1997).
Author Response
Dear reviewer,
Re: Manuscript ID: molecules-1883212 and Title: Research progress on the mechanisms of polysaccharides against gastric cancer.
Thank you very much for your comments and professional advice. These ideas help to enhance the academic rigor of our article. According to your suggestion and request, we have corrected the revised manuscript. We hope that our work can be improved again. In addition, we want to show the following details.
Reviewer #1
1. What stands out is that PSK, lentinan and laminaran, all derived from different biological sources, are beta glucans having 1-6 and 1-3 linkages. Many of the other anti-cancer polysaccharides described in the review are alpha glucans. This suggests that one possible origin of the immunomodulatory activity of glucans may be defense against mycobacteria, since the surface integument of M. bovis BCG consists largely of an alpha glucan that may account for most of BCG's anti-cancer activity, possibly as a T-cell stimulant (1-3).
Authors' Reply: We thank you for pointing this out. We add these on page 14. The details are as follows: "Since the 1930s, studies of patients with gastric cancer, melanoma and leukaemia treated with BCG have shown remission or non-recurrent disease [148]. In addition, it is worth noting, PSK, 菭 mushroom and layer adhesion, all creatures from different sources, is to have 1-6 and 1-3 keys of beta glucan, and review described in many other anti-cancer polysaccharides is alpha glucan. This suggests that a possible origin of glucan immunomodulatory activity may be defense against mycobacteria, since the surface outer coat of Mycoplasma BCG consists mainly of αglucan, which may account for the majority of BCG's anticancer activity and may act as a T-cell stimulant [149-151].
2. The authors should re-examine the revised manuscript for English errors, since I noticed several misspellings and syntactical errors in the revised text.
Authors' Reply: This was an oversight on our part and we have corrected the problem in the manuscript.
We would like to thank you again for taking the time to review our manuscript. We look forward to hearing from you.
Sincerely yours,
School of Food and Biological Engineering, Xihua University
Liping Chen lpingchen2022@163.com (L.C.)
Ling Li 0120030074@mail.xhu.edu.cn (L.L.)
Tel. / Fax: +86 15184471605.
5 Sep. 2022
Reviewer 2 Report
I accept introduced correction, however I am still not convinced of the title of the publication, is it spelled correctly in terms of grammar?
Please, check how to write Radix Astragali. It is not the name of the species but the plant material.
Author Response
Dear reviewers,
Re: Manuscript ID: molecules-1883212 and Title: Research progress on the mechanisms of polysaccharides against gastric cancer.
Thank you very much for your comments and professional advice. These opinions help to improve academic rigor of our article. Based on your suggestion and request, we have made corrected modifications on the revised manuscript. We hope that our work can be improved again. Furthermore, we would like to show the details as follows.
Reviewer·2#
1.Introduction“I accept introduced correction, however I am still not convinced of the title of the publication, is it spelled correctly in terms of grammar?”
The auther’ s answer:It was our negligencet, and we have corrected the title which is “ Research progress on the mechanisms of polysaccharides against gastric cancer.”
2. Introduction“Please, check how to write Radix Astragali. It is not the name of the species but the plant material.”
The auther’ s answer:We thank to you for pointing this out, and we have found that it should be written as Astragalus membranaceus, so we have corrected it in manuscript.
We would like to thank you again for taking the time to review our manuscript.We look forward to hearing from you.
Sincerely yours,
School of Food and Biological Engineering, Xihua University
Liping Chen lpingchen2022@163.com (L.C.)
Ling Li 0120030074@mail.xhu.edu.cn (L.L.)
Tel. / Fax: +86 15184471605.
5 Sep., 2022